# A Novel Inlet for Enriching Concentrations of Reactive Organic Gases in Low Sampling Flows

Namrata Shanmukh Panji[1] and Gabriel Isaacman-VanWertz[1]

[1]Department of Civil and Environmental Engineering, Virginia Tech, Blacksburg, VA, 24061, USA

**Correspondence:** Namrata Shanmukh Panji (namratapanji@vt.edu) and Gabriel Isaacman-VanWertz (ivw@vt.edu)

**Abstract.** Preconcentration of samples is often necessary to detect the low levels of volatile organic compounds present in the atmosphere. We introduce a novel inlet that uses selective permeation to continuously concentrate organic gases in small sample flows (up to several sccm), and consequently improve the sensitivity and limits of detection of analytical instruments. We establish the dependence of enrichment on the sample flow (decreasing with increasing flow) and pressure differential across its walls (increasing with increasing pressure differential). We further show that while there is some dependence on the permeability of the target analyte, most analytes of atmospheric interest exhibit similar enrichment. Enrichments between 4640% and 111% were measured at flows of 0.2 to 3 sccm for major reactive atmospheric gases: isoprene ($C_5H_8$), monoterpenes ($\alpha$-pinene, $C_{10}H_{16}$), and alkanes ($C_3$-$C_6$). The relationship between inlet design parameters, operating conditions, and inlet efficiency, are modeled and validated, enabling predictable enrichment of most atmospheric gases.

## 1 Introduction

A wide range of volatile organic compounds (VOCs) are emitted through natural processes (biogenic emissions) and by human-made processes (anthropogenic emissions). Once emitted, these compounds undergo photochemical oxidation to produce secondary organic aerosols (Kroll and Seinfeld, 2008) or produce ozone in the presence of NO and $NO_2$ ($NO_x$) (Haagen-Smit, 1952). Despite their sizeable emissions (∼1300 Tg C/year (Goldstein and Galbally, 2007)), they are often present in mixing ratios ranging from a few parts per trillion (ppt) to parts per billion (ppb) and have large impacts on human health, radiative forcing, and air quality (Ebi and McGregor, 2008).

As the atmosphere is a complex mixture of thousands of organic gases (Goldstein and Galbally, 2007), analyses of its constituents presents several challenges. These organic compounds span wide ranges in volatilities (and hence phases), lifetimes, and mixing ratios, and require highly sensitive instrumentation to be detected. To that end, complex and often expensive instrumentation such as gas chromatography (GC) coupled to a suitable detector, such as, a mass spectrometer (MS), flame ionization detector (FID), flame photometric detector (FPD), or electron capture detector (ECD), are used. On the other hand, recent developments in low-cost VOC sensors have prompted an increase in their application to measure total VOCs. Spinelle et al. (2017) reviewed several low-cost VOC sensors and note that certain photo ionization detectors (PID), portable GCs, and e-nose technologies offer limits of detection in the ppb range. Despite this, when it comes to measuring certain VOCs in the ppt to sub-ppb range, sensitivities of low-cost sensors fall short. Consequently, although many of the instruments currently used

offer sensitivities in the ppb range, preconcentration steps are required to address the VOC levels in ambient samples below these limits of detection (Michulec et al., 2005).

Preconcentration of VOCs for both active and passive sampling is achieved in several ways (Ras et al., 2009). Cryogenic trapping, which involves passing the sample through a cold tube with glass beads or silica granules (Bourtsoukidis et al., 2017), is a well-established method used for preconcentration of VOCs in atmospheric samples but presents issues with ice forma-tions clogging sample flows (Oliver et al., 1996; Wang and Austin, 2006). Adsorptive enrichment using solid sorbents such as Tenax, Carbopack, Porapack Q, etc., is also employed extensively but typically requires thermal desorption to retrieve the VOCs (Demeestere et al., 2007). Similarly, carbon nanotubes (CNTs) are being increasingly studied as preconcentrators (Li et al., 2004; Duran et al., 2009; Hussain et al., 2008), but require the same multi-step purge-and-trap configurations as other sorbent based technologies and may not yield uniform preconcentration (Hussain and Mitra, 2011). As for preconcentration for on-line VOC detection, membrane introduction mass spectrometers (MIMS) are currently being employed and improved upon (Ras et al., 2009). However, techniques such as the MIMS are rather complete instruments which contain a component that pre-concentrates sample flows. By introducing a simple inlet as in our work, we plan to lay the groundwork for a preconcentration technique adaptable with other detectors such as those compatible with GCs.

In an effort to provide preconcentration of atmospheric samples with compounds of various volatilities for on-line sampling, we present a novel enriching inlet that uses a semipermeable Teflon$^{TM}$ AF-2400 membrane. This membrane has previously been used as a continuous flow reactor to facilitate reactions at gas-liquid boundaries and liquid-liquid boundaries (Polyzos et al., 2011; Skowerski et al., 2014; O'Brien et al., 2011; Mastronardi et al., 2013) but its application for the continuous enrichment of atmospheric VOC samples remains unexplored. The following section provides the operating mechanism of the enriching inlet in greater detail.

## 2 Methods

### 2.1 Principle of Operation

The steady state flux of a gas through a permeable membrane is given by equation (1),

$$J_i = P_i \times \frac{p_{high} - p_{low}}{x} = P_i \times \frac{\Delta p}{x} \tag{1}$$

where $J_i$ is the steady state gas flux of some component, $i$, of the gas (typically in cm$^3$(STP) cm$^{-2}$ s$^{-1}$), $p_{high}$ and $p_{low}$ are the pressures on the high and low pressure sides of the membrane respectively in cmHg (or $\Delta p$ is the pressure difference across the wall of the permeable membrane), $x$ is the thickness of the permeable membrane in cm, and $P$ is the permeability coefficient of the gaseous species (Pinnau and Toy, 1996). $P$ is commonly expressed in terms of the non-SI unit Barrer, equal to $10^{-10}$ cm$^3$(STP) cm cm$^{-2}$ s$^{-1}$ cmHg$^{-1}$ or $3.35 \times 10^{-11}$ mol m m$^{-2}$ s$^{-1}$ bar$^{-1}$.

The gas flux $J$ can be expressed in terms of volumetric flowrate that permeates across the membrane using Equation (2) where $Q_{perm,i}$ is in standard cubic centimeters per minute, sccm or cm$^3$(STP) min$^{-1}$ and $A$ is the surface area of the permeable

membrane in cm$^2$.

$$Q_{perm,i} = J_i \times A \tag{2}$$

In this work, we exploit this permeability with the purpose of enriching ambient sample flows to have higher concentrations of trace gases of interest. Air is sampled through tubing comprised of a permeable membrane subjected to a pressure differential. In all work shown here, sample flow through the tubing is maintained at a higher pressure, allowing permeation of gases out of the sample, as shown in Figure 1, though in principle the system could be designed to operate under a reversed pressure differential.

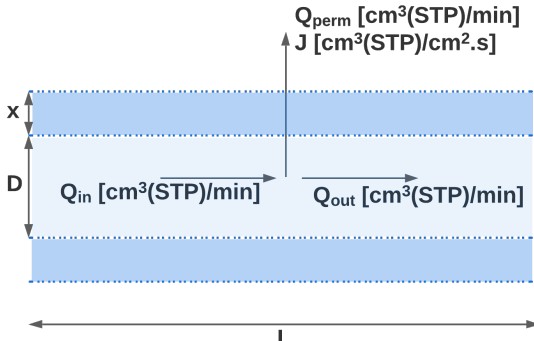

**Figure 1.** Longitudinal cross-section of the Teflon$^{TM}$ AF-2400 inlet (represented by the blue dashed regions) of length $L$, internal diameter of $D$, and wall thickness of $x$. The volumetric flow rates entering the tubing, moving across the tubing walls, and leaving the tubing are denoted by $Q_{in}$, $Q_{perm}$, and $Q_{out}$ respectively. The volumetric flux of the gaseous species moving across the tubing is denoted by $J$.

Under these conditions, $A$ in Equation (2) is given by $\pi DL$ where $D$ and $L$ are the inner diameter and length of tubing respectively. In this work, we use Teflon$^{TM}$ AF-2400, a commercially available amorphous glassy copolymer of tetrafluoroethylene (TFE) and 2,2-bis(trifluoromethyl)-4,5-difluoro-1,3-dioxole (BDD). The gas permeability through Teflon$^{TM}$ AF-2400 follows a size-sieving trend where permeability decreases with increase in critical volume of the gaseous species transporting across the membrane (Zhang and Weber, 2011). In principle, the approach described here could be applied to other permeable materials, provided the material is selectively permeable between analytes of interest and air, but Teflon$^{TM}$ AF-2400 affords one of the highest currently known permeabilities for air. Furthermore, Teflon is a preferred material for sampling atmospheric gases (Deming et al., 2019), so it is an ideal material for this application.

Combining equations (1) and (2) and accounting for the unit conversions yields Equation (3):

$$Q_{perm,i}\,[sccm] = 60 \left[\frac{s}{min}\right] \times 75 \left[\frac{cmHg}{bar}\right] \times 10^{-10} \times P_i\,[Barrer] \times \frac{\Delta p\,[bar] \times A\,[cm^2]}{x\,[cm]} \tag{3}$$

The standard volumetric permeation rate can be understood as a mass or molar flow as in Equation (4) :

$$Q_{perm,i}\left[\frac{moles}{min}\right] = 60\left[\frac{s}{min}\right] \times 3.35 \times 10^{-11} \times P_i\,[Barrer] \times \frac{\Delta p\,[bar] \times A\,[m^2]}{x\,[m]} = \frac{0.335}{100\left[\frac{cm}{m}\right] \times 75\left[\frac{cmHg}{bar}\right]} \times Q_{perm,i}\,[sccm]$$

(4)

In this work, the volumetric flowrate leaving the inlet as the sample flow ($Q_{out}$ in Figure 1) is a preset condition, set by the sampling instrument. Flow into the inlet is simply the sum of permeation flow and outlet flow, as in Equation (5) as:

$$Q_{in,i} = Q_{perm,i} + Q_{out,i} \tag{5}$$

As the goal of this work is to sample trace gases in a sample flow dominated by air, we can consider that a given total flow, $Q_{j,tot}$ is the sum of the combined flow rates of air and any other components of the gas (e.g., an analyte of interest), and for dilute gases in a predominantly air sample flow is approximately equal to the air flowrate:

$$Q_{j,tot} = Q_{j,air} + Q_{j,analyte} + ... \sim Q_{j,air} \tag{6}$$

Each analyte in each flow, $j$, exists with some concentration, $C$, measured as its mixing ratio (moles of analyte per mole of air):

$$C_{j,analyte} = \frac{Q_{j,analyte}}{Q_{j,air}} \tag{7}$$

Each analyte will have a permeability across the membrane according to their critical volume, temperature, interactions with the tubing polymer, and to some extent pressure differentials (Pinnau and Toy, 1996). In the case of an analyte that has lower permeability than air, relatively more air will cross the membrane, and the amount of analyte in $Q_{out}$, will be enriched relative to the remaining air. This enrichment can be quantified as the excess concentration analyte in the outlet flow relative to the inlet flow.

$$Enrichment = \frac{C_{out,analyte}}{C_{in,analyte}} - 1 = \frac{\frac{Q_{out,analyte}}{Q_{out,air}}}{\frac{Q_{in,analyte}}{Q_{in,air}}} - 1 \tag{8}$$

The maximum possible concentration enrichment occurs for any analyte that has no permeability across the membrane. In such a case, no analyte is lost by permeation and $Q_{in,analyte}$ is equal to $Q_{out,analyte}$. Enrichment at this extreme condition of no analyte permeability is given as:

$$Maximum\ enrichment = \frac{\frac{Q_{in,analyte}}{Q_{out,air}}}{\frac{Q_{in,analyte}}{Q_{in,air}}} - 1 = \frac{Q_{in,air}}{Q_{out,air}} - 1 \tag{9}$$

From Equations (8) and (9), enrichment is primarily determined by the amount of air that permeates across the membrane. The maximum enrichment is simply the ratio of how much air enters the inlet versus how much flows to the instrument, and all analyte mass within the permeated flow remains in the minor outlet flow.

In reality, all analyte gases can permeate across the membrane to some degree. The degree to which one component crosses the membrane relative to another (the "selectivity") is a function of the ratio of permeabilities. The selectivity of interest

in this work is that of the analyte (which ideally remains in the sample flow) relative to air (which is removed across the membrane), $P_{analyte}/P_{air}$. At the extreme condition of no selectivity, where analyte permeability approaches that of nitrogen and/or oxygen, the composition of the permeation flow is identical to that of the sample flow and there is no enrichment. At intermediate selectivity, $Q_{perm}$ is depleted in analyte relative to air, with remaining analyte mass concentrated into the sample

flow. A general form describes extreme and intermediate cases, in which enrichment of an analyte is reduced from maximum enrichment as a function of the selectivity:

$$Enrichment[\%] = (\frac{Q_{in,air}}{Q_{out,air}} - 1) \times (1 - \frac{P_{analyte}}{P_{air}}) \times 100 \tag{10}$$

Total enrichment can therefore be calculated by combining Equations (3), (5), and (10) as:

$$Enrichment[\%] = (P_{air} - P_{analyte}) \times \frac{\Delta p\,[bar]}{Q_{out,air}\,[sccm]} \times \frac{A\,[cm^2]}{x\,[cm]} \times 60\left[\frac{s}{min}\right] \times 75\left[\frac{cmHg}{bar}\right] \times 10^{-10} \times 100 \tag{11}$$

In Equation (11), the first term represents the selectivity of the analyte and membrane material, the second term represents system operating conditions, the third term represents physical parameters of the tubing membrane, and the remaining terms represent unit conversions and constants. For the purpose of our application, we note that permeabilities of most analytes of interest are an order of magnitude lower than that of air (Alentiev et al., 2002). Permeabilities of mixed gases typically diverge from pure gas permeabilities or ideal mixing and vary by temperature and pressure differential. Furthermore, air is itself a

mixture of two gases, so while an average permeability of nitrogen and oxygen (~600 Barrer for room-temperature Teflon[TM] AF-2400 (Alentiev et al., 2002)) is used in this work to describe air, this approach may not exactly represent the permeability of the mixture. It is therefore generally expected that a precise permeability or selectivity will not be known for each analyte in the complex atmospheric mixture. However, most analytes in the atmosphere are sufficiently large to maximize selectivity in the material used in this work and are shown to approach maximum enrichment, so a precise accounting of permeability is

likely not necessary.

The following section describes the experimental method used to quantify the extent of enrichment for selected compounds of interest. In subsequent discussion, $Q_{out}$ is referred to as $Q$ for simplicity, as this is the flow that is actually sampled by the detector downstream.

## 2.2 Experimental Setup and Data Analysis

*High concentration experiments*. Experiments were conducted to evaluate the performance of the enriching inlet using the setup as shown in Figure 2. Pressurized gas cylinders were used to deliver the sample to the enriching inlet through a pressure regulator at pressures between 1 and 3 bar absolute (14.5 to 43.5 psia) and tests were performed at 0.2 sccm to 5 sccm. These low flows are aligned with typical GC flows and an application for future research is expected to incorporate this approach into a combined instrument that would allow some separation of analytes in ambient atmospheres provided some engineering

and analytical challenges could be overcome. Portable calibration gas cylinders were used for methane (1% Methane Balance Air Certified Standard Mixture, Airgas®), propane (0.6% Propane Balance Air Certified Standard Mixture, Airgas®), and butane (0.9% N Butane Balance Air Certified Standard Mixture, Airgas®). Samples for isoprene, $\alpha$-pinene, cyclohexane, and

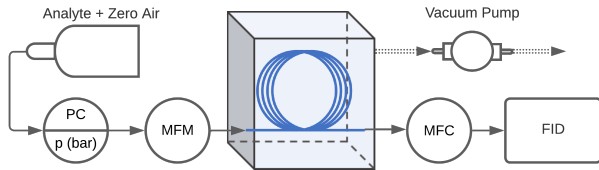

**Figure 2.** Schematic of the experimental setup used to measure enrichment offered by the inlet. The Teflon[TM] AF-2400 is represented by the blue line, the pressure controller is denoted by PC, the mass flow controllers upstream and downstream of the enrichment inlet are denoted by MFM (mass flow meter, used for measurement only) and MFC (mass flow controller) respectively, and the flame ionization detector is denoted by FID

n-pentane were acquired at >98% purity from Sigma-Aldrich and prepared in 6 litre TO-Can Air Sampling Canisters (Restek Corporation). Ultra Zero Grade Air (Airgas®) was used as a make up gas in the canisters. The approximate concentrations of the samples in the canisters were 600 ppm for $\alpha$-pinene and 900 ppm for isoprene, cyclohexane, and n-pentane. The pressure downstream of the cylinder or canister was controlled using a pressure regulator (R00-01-000, Wilkerson Corporation). The sample was delivered to the Teflon[TM] AF-2400 tubing (Biogeneral Inc.), which had an internal diameter of 0.61 mm, wall thickness of 0.064 mm, and length of 325 cm. Two mass flow controllers (Alicat Scientific, Inc.) were used to measure flow into the tubing, $Q_{in}$, (MCS-100sccm, MFM in Figure 2) and control flow to the detector $Q_{out}$ (MCS-10sccm, MFC in Figure 2). Accuracy of flows is reported by the manufacturer as 0.2% of full scale plus 0.8% of reading; most flows in this work are measured at 2-20% of full scale, indicating uncertainty of less than 10% even at low reported flow rates. The enriching inlet rested in an airtight aluminum container (approximately 15cm x 10cm x 7.5cm) that was either open to the atmosphere (high pressure testing conditions) or closed and reduced to 0.2 bar absolute (sub-atmospheric testing conditions) by connecting to a diaphragm vacuum pump (GAST Manufacturing). A Flame Ionization Detector (FID, SRI Instruments) was used to detect the analyte concentration. This detector was chosen due to its availability, robustness, and flexibly engineered design for the low flows used in this work. The minimum detection limit of this detector is 1 ppm, requiring sample concentrations in the ppm to percent levels for testing the enriching inlet. The FID signal was logged using an Arduino Uno R3 (Arduino Corporation, Somerville, MA) where the readings were averaged every second as they were being recorded.

*Low concentration experiments.* A small number of experiments were performed at concentrations approaching atmospheric levels in order to evaluate whether observed enrichment is concentration independent and can be achieved at relevant concentrations. Experiments were performed at sub-ppm concentrations using a slightly different experimental setup. A ppb miniPID2 photoionization detector (ION Science Inc.) was used in place of the FID to detect the analyte. Analyte samples were prepared in a 1 Liter Tedlar®bag (Restek Corporation) using 0.5 ppm Benzene Calibration Gas (GASCO Affiliates, LLC.) with balance Ultra Zero Grade Air (Airgas®). The enrichment tests were carried out with a 300 cm length of enriching inlet at 0.72 bar sub-atmospheric pressure differential and a flow rate of 2 sccm. The experimental set-up was as shown in Figure B1. The

enrichment tests were conducted in triplicate. An exponential model was used to fit to the 1 second resolution data to quantify the stable equilibrium concentration at enriched and unenriched conditions.

### 2.2.1 Calibration of the FID

Prior to each experiment, the response of the FID (in volts) was measured as a function of analyte mass by varying the mass flow to the detector. To keep any losses to instrument lines constant, only the enriching inlet was removed. A polynomial fit was used to relate the FID response and mass flow to account for any non-linearity in the FID response, though response was typically linear. The $R^2$ value of all calibration curves were above 0.98, and in all cases (unless otherwise denoted in Table 1) FID response was calibrated across the entire range of enriched and unenriched concentrations being measured (i.e., calibration curves were not extrapolated).

### 2.2.2 Experimental conditions

Experiments were conducted to study the relationships between enrichment and operating conditions. For each experiment, sample delivery pressure was set using the regulator (PC, Figure 2) and the same supply pressure was used for both calibration (i.e., FID response to variable flow rates without the inline enriching tubing) and enrichment experiments (i.e., with the inline enriching tubing). Sample flowrate to the detector was set on the MFC and enriched FID response allowed to stabilize based on visual inspection (roughly <5% change over several minutes). Inlet flow measured using the MFM was compared to the expected permeation flow estimated from Equation (4) and generally found to agree; in cases of non-agreement, the experimental setup was examined for leaks and the experiment repeated if a leak was found. Example data in Figure 3 demonstrates FID response for propane at a pressure differential of 3 bar and sample flowrate of 0.5 sccm.

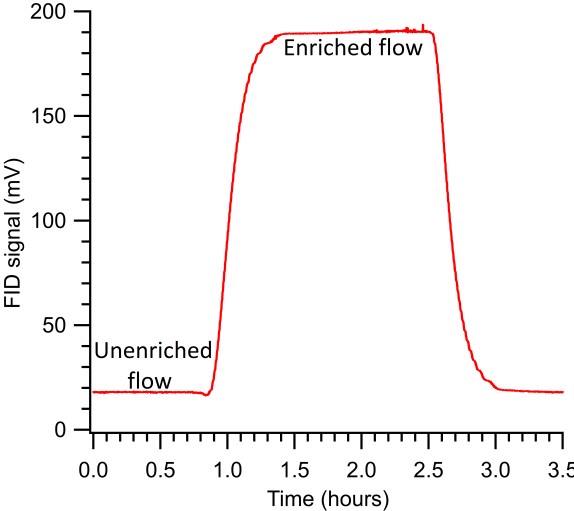

**Figure 3.** FID signal measured during a sample experiment run studying enrichment for propane at $\Delta P = 3$ bar (SA), $Q = 0.5$ sccm, and enriching tubing length of 325 cm. The run was started at roughly the 0.8 hour mark and stopped at 2.50 hour mark.

For each experiment, enrichment is calculated as in equation (12) from the measured concentration (i.e., calibrated FID response) of the analyte during enrichment, $C_{enriched}$, relative to the concentration in the sample mixture. The concentration in the sample mixture is the baseline concentration, $C_{baseline}$, representing unenriched flow at the set sample flowrate. The enriching tubing was disconnected after each experiment and any remaining sample in the tubing was removed using a pump, before repeating or conducting a new experiment. The average and standard deviation of enrichment over triplicate experiments was calculated for each set of operating conditions and is reported in Table A1.

$$Enrichment\,(\%) = \frac{C_{enriched} - C_{baseline}}{C_{baseline}} \times 100 \tag{12}$$

In addition to validating the prediction of enrichment based on the operating principles described above, experiments were designed to understand the effect of several operating conditions, particularly sample flowrate at the detector ($Q$) and pressure differential across the membrane ($\Delta P$). Furthermore, results at a pressure differential $\Delta P$ are dependent on the absolute pressures used to achieve the differential. The pressure downstream of the permeable membrane affects the permeation rate (Alentiev et al., 1997; Merkel et al., 1999; Alentiev et al., 2002), so analyte permeation rates are expected to be lower through the membrane when the downstream pressure is close to vacuum (sub-atmospheric operation). Based on permeability coefficients available in literature for low downstream pressures (Alentiev et al., 1997) compared to high pressure conditions (Alentiev et al., 2002), changes in permeation rates under low downstream pressures are estimated to improve enrichment by $\sim$20% for methane. However, while a trend of improved enrichment under sub-atmospheric conditions is expected, quantitative estimation of improvement (including this 20% estimate) is highly uncertain due to a lack of data on most analytes and differences between specific polymers. We therefore examine pressure differentials in two ways: (1) high pressure (HP) operating condition where the sample flow inside the enriching tubing was maintained at a high pressure and the outside of the tubing at ambient pressures, and (2) sub-atmospheric (SA) operating condition where the sample flow inside the enriching tubing was maintained at high or ambient pressures and the outside of the tubing at vacuum pressures. We note that ambient-pressure sampling is limited to a 1 bar pressure differential, but many analytes of interest can be passed through a compressor or pump without significant losses (Lerner et al., 2017), suggesting that a pressurized sample stream could enable high enrichments. Enrichment was further studied using different analytes (methane, propane, butane, pentane, cyclohexane, $\alpha$-pinene, and isoprene) to study the influence of the permeability of the analyte molecule on sample enrichment. A list of all conditions studied is presented in Table 1. Each condition was tested in triplicate unless otherwise noted. For all experiments shown, the physical dimensions of the enriching tubing used were: $D = 0.61$ mm, $x = 0.064$ mm, and $L = 325$ cm. Residence time in the enrichment tubing with these physical dimensions is approximately 1 minute at 1 sccm (and scales linearly with flow rate). In addition, tests were carried out to verify the mutual independence of enrichment and analyte concentration and have been detailed in Appendix B.

**Table 1.** List of operating conditions tested to study enrichment offered by a 325 cm length of enriching inlet

| Analyte | Permeability coefficient, $P$ (Barrer) | Flow, Q (sccm) | Pressure differential, $\Delta P$ (bar) | Pressure condition |
|---|---|---|---|---|
| methane | 400 | 2 | 2 | SA, HP |
| butane | 20 | 2 | 2 | SA, HP |
| pentane | 35 | 2 | 2 | SA, HP |
| $\alpha$-pinene | 8 | 2 | 2 | SA, HP |
| isoprene | 40 | 2 | 2 | SA, HP |
| cyclohexane | 10 | 0.2 | 3 | SA |
|  |  | 0.5 | 3 | SA |
|  |  | 1 | 2 | SA, HP |
|  |  | 1 | 3 | SA |
|  |  | 2 | 2 | SA, HP |
|  |  | 3 | 1 | SA, HP |
|  |  | 5 | 3 | SA[†] |
| propane | 100 | 0.2 | 3 | SA |
|  |  | 0.5 | 3 | SA, HP[*] |
|  |  | 1 | 1 | SA, HP |
|  |  | 1 | 2 | SA, HP |
|  |  | 1 | 3 | SA, HP[*] |
|  |  | 2 | 2 | SA, HP |
|  |  | 5 | 3 | SA[†] |

SA: Sub-atmospheric condition; HP: High pressure conditions; [*] averaged over n=2; [†] required extrapolation for calibration.

# 3 Results and Discussions

## 3.1 Effect of Operating Conditions

From the operating principle of the permeable membrane, the amount of air permeating out of the enriching inlet is independent of the sampling flowrate to the detector (see Equation 3 and 4). Instead, increasing this flowrate to the detector increases only the amount of sample being pulled into the enriching inlet by that amount (see Equation 5). Consequently, with increasing flowrate, the permeation remains constant and the volume of flow to the detector into which it is concentrated increases, so enrichment is expected to reduce (see Equation 10). To examine this relationship, the enrichment factor of propane and

cyclohexane was characterised over a range of flowrates and observed to decrease with increasing flowrate as expected (Figure 4). Furthermore, observed enrichment tends to somewhat exceed that predicted for a given flowrate and pressure differential (dashed lines), which may be due to uncertainty in the estimated permeability coefficient or may be due to complex permeation behavior caused by the inherent pressure drop along the length of the permeable tubing, which acts as a restriction due to its narrow diameter. Deviations between modeled and observed enrichment are likely due in part to uncertainty in permeabilities, as literature values were measured at slightly different temperatures and pressures (Alentiev et al., 2002) than those used here and it is difficult to quantify the precise impact of these differences on permeability for most of the analytes measured. In addition, the relationship with critical volume used to estimate permeability introduces some uncertainty. Maximum enrichment of 3050±56% and 4640±414% (i.e., factor of 31.5 and 47.4 times) was achieved for propane and cyclohexane samples, respectively at the lowest flowrate of 0.2 sccm and the highest pressure differential, of 3 bar. We also note that the response time of the system can be reduced to a few minutes in contrast to the long response time observed in Figure 3 by temporarily increasing sampling flow rates (e.g., 8 sccm purge flow, see Figure B2).

Since permeation rate through the membrane is directly related to the difference in pressure across the thickness of its walls, enrichment is expected to increase as the pressure difference increases. Examined as a function of pressure differential, the same data shown in Figure 4 demonstrate the expected increase in enrichment with increasing pressure difference (Figure 5) for propane and cyclohexane. In most cases, enrichment is slightly higher when the pressure differential is achieved through sub-atmospheric operation (i.e., vacuum is maintained outside of the enriching inlet carrying a sample flow at atmospheric pressure or higher), likely due to the pressure dependence of permeability coefficients (Alentiev et al., 2002).

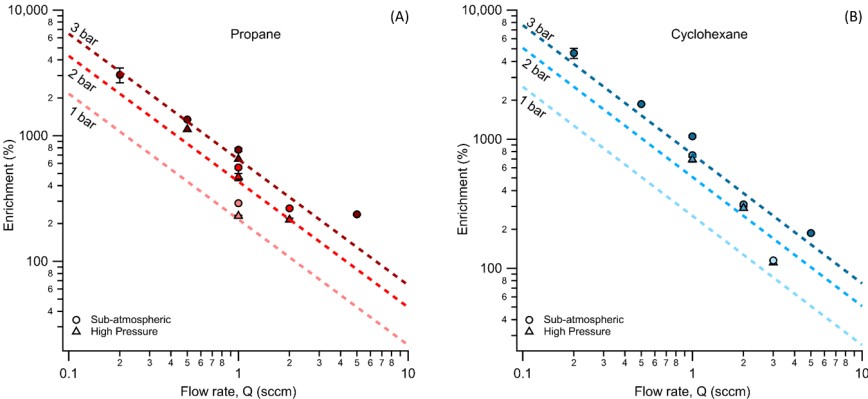

**Figure 4.** Average measured enrichment (error bars of $1\sigma$ over repeated trials) for (A) propane and (B) cyclohexane at sampling flowrates of 0.2, 0.5, 1, 2, and 5 sccm. Dashed lines represent the theoretical enrichment estimated for the operating conditions tested where increasing pressure differential is distinguished by a darkening color gradient. The different modes of testing, sub-atmospheric and high pressure, are denoted by circular and triangular markers respectively.

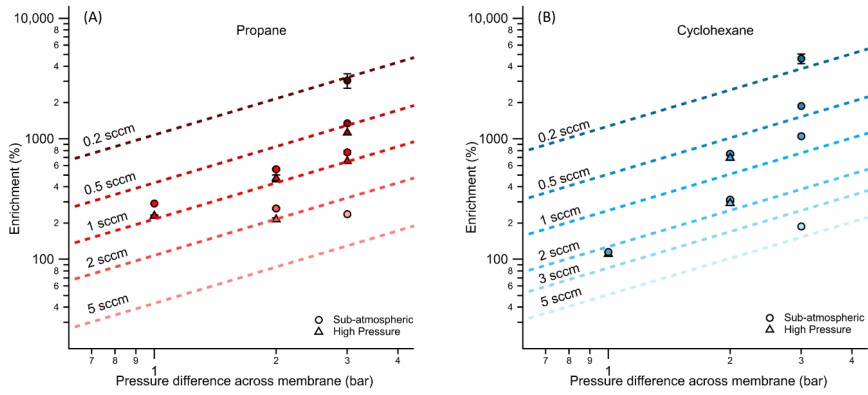

**Figure 5.** Average measured enrichment (error bars of $1\sigma$ over repeated trials) for for (A) propane and (B) cyclohexane at pressure differential of 1, 2, and 3 bar across the walls of the enriching inlet. Dashed lines represent the theoretical enrichment estimated for the operating conditions tested where decreasing sampling flowrate is distinguished by a darkening color gradient. The different modes of testing, sub-atmospheric and high pressure, are denoted by circular and triangular markers respectively.

## 3.2 Effect of Analyte Permeability

As shown in Equation 11, enrichment depends on the difference in permeability coefficients between air and an analyte of interest. The permeability coefficient of any gaseous species is a function of critical volume (Alentiev et al., 2002), so to examine the effect of permeability on enrichment, a range of analytes with varying critical volumes were measured under the same set of operating conditions (pressure differential of 2 bar, sampling flowrate of 2 sccm). Critical volumes were obtained using the UManSysProp (Topping et al., 2016) critical property predictive tool which employs the estimation methods described

by Nannoolal et al. (2004). Methane, with a critical volume of 114.94 cm$^3$/mol, has been measured to have a permeability coefficient through Teflon$^{\text{TM}}$ AF-2400 roughly 80% of nitrogen at 25°C at 50 psig feed pressure while propane was measured to be 25% as permeable as nitrogen at the same conditions (Pinnau and Toy, 1996). Hence, the permeabilities of compounds with critical volumes similar to and above that of propane would be low. Consequently, although permeabilities and thus selectivities are sensitive to feed pressure and other factors (Alentiev et al., 2002), most compounds of atmospheric interest

can be expected to exhibit similar (and nearly maximum) enrichment. In contrast, compounds with critical volumes close to that of air (84.5 cm$^3$/mol (Lemmon et al., 2000)) can be expected to roughly permeate across the walls of the enriching inlet to the same degree as air, and experience lower enrichment. This effect is reflected in our results (Figure 6) where the enrichment of methane (CH$_4$) was 70$\pm$3% and 40$\pm$1% for sub-atmospheric and high pressure operating condition respectively, while the enrichments for propane, n-butane, n-pentane, cyclohexane, isoprene and $\alpha$-pinene were 163% to 312% with only a minor trend

with molecule size. In general agreement with the size-dependant sieving manner of permeation, the enrichment for $\alpha$-pinene is the highest, while enrichments of propane, n-butane, n-pentane and isoprene are only somewhat lower and approximately agree with the expected range of enrichment. Small differences in enrichments could be due to chemical interactions between the analyte molecules and the membrane.

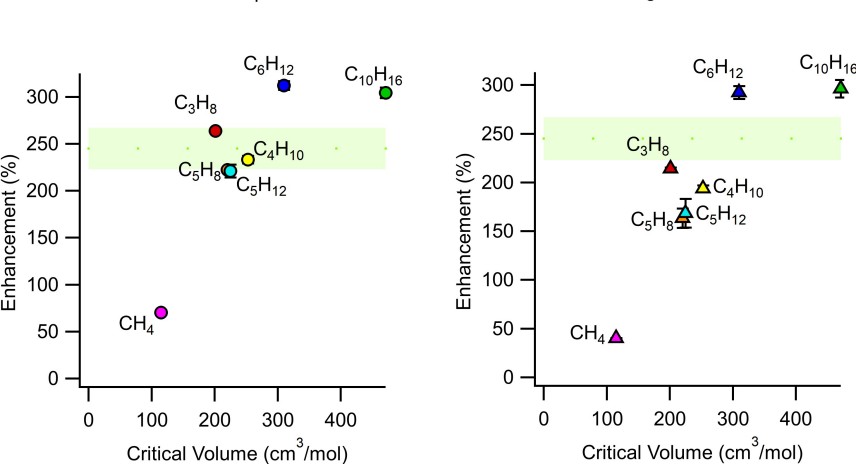

**Figure 6.** Average measured enrichment (error bars of $1\sigma$ over repeated trials) as a function of critical volume for methane ($CH_4$), propane ($C_3H_8$), butane ($C_4H_{10}$), pentane ($C_5H_{12}$), cyclohexane ($C_6H_{12}$), isoprene ($C_5H_8$), and $\alpha$-pinene ($C_{10}H_{16}$) at (A) sub-atmospheric, and (B) high pressure conditions. These tests were conducted at a pressure differential of 2 bar across the walls of the enriching inlet, sampling flowrate of 2 sccm using an enriching inlet 325 cm in length. The green shaded area represents the expected enrichment for analytes with a permeability coefficient in the range of 0 to 100 Barrer, within which most environmental compounds are expected to lie.

### 3.3 Effect of Analyte Concentration

Since flame ionization detectors are sensitive to compounds only at concentrations as low as hundreds of ppm, previously discussed experiments were conducted at concentrations well above typical ambient conditions. In order to ensure that enrichment can be achieved at atmospherically relevant conditions, a subset of experiments were performed at sub-ppm concentrations of benzene (as low as 45 ppb) using a photionization detector. The effect of analyte concentration on enrichment was quantified using the setup shown in Figure B1. The averages of triplicate measurements at three different analyte concentrations: 45, 125, and 500 ppb of benzene ($C_6H_6$) are as reported in Figure 7. We observe that the concentration of the analyte does not significantly affect the level of enrichment. Though some apparent effect is observed, measurements at lower concentrations here are near this instrument's limit of detection and subject to some measurement uncertainty. Concentrations in these experiments are orders of magnitude lower than those in other experiments but still approximately follow predicted values even at these low concentrations, suggesting little impact of concentration if any. This, combined with the fact that residence times can be reduced by temporarily using high sampling flows (refer to Figure B2), suggests that transient changes in the atmospheric concentrations would be detectable with the enriching inlet but it would need to be tested in real-world ambient environments. Once further advancements are made to transform the preconcentration inlet into a fully operational instrument, tests can be conducted under real-life conditions to assess its performance.

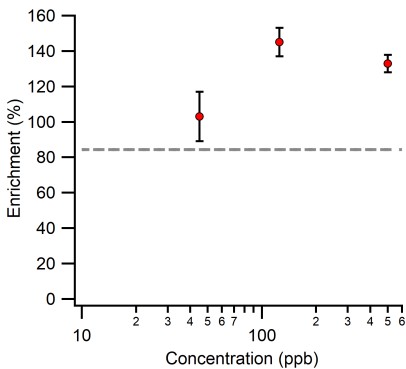

**Figure 7.** Average measured enrichment (error bars of $1\sigma$ over repeated trials) as a function of analyte concentration at 45, 125, and 500 ppb. The tests were performed using an enriching inlet that was 300 cm long, with a pressure differential of 0.72 bar across its walls and a sampling flowrate of 2 sccm. The gray dashed line represents the theoretical enrichment expected for benzene ($C_6H_6$) with a permeability coefficient of 20 Barrer.

## 4   Conclusions

We have proposed and tested a novel method for the preconcentration of VOCs in atmospheric samples. It was tested at different sampling flowrates, and pressure differentials across the membrane walls for compounds with a broad range of critical volumes. The enrichment factor was found to be roughly predictable using the set of equations developed in this paper. We demonstrated enrichments greater than 3050% and 4640% for propane and cyclohexane at a sampling flowrate of 0.2 sccm. Enrichments of 163%-312% for propane, n-butane, n-pentane, cyclohexane, $\alpha$-pinene, isoprene were achieved at sampling

flowrates of 2 sccm, suggesting that enrichment is possible and approximately similar for compounds across a wide range of atmospherically relevant compounds. Even at a moderate pressure difference of 1 bar under sub-atmospheric conditions, enrichments of several hundred percent were measured for propane and cyclohexane at flows between 1 and 3 sccm, suggesting that the inlet is capable of enriching ambient samples without the need for a pressurized sample flow (though doing so would increase enrichment). By decreasing sample flowrate or using longer lengths of the enriching inlet, higher levels of enrichment

could be achieved, implying potential significant improvements (factors of 2 or higher) in instrument sensitivity for instruments capable of operating at sccm-level flowrates. Furthermore, due to the temperature dependence of permeation through Teflon$^{TM}$ AF-2400, there may be additional opportunities to modulate operating conditions to improve enrichment, for instance by heating the enriching inlet. However, temperature impacts permeabilities of both air and an analyte of interest, leading to a complex impact on enrichment that is not well theoretically constrained due to a lack of data on the temperature dependence of

analyte permeation (Merkel et al., 1999). The magnitude of this enhancement is generally not expected to be large, and ambient temperature cycles are not expected to drive significant changes in enrichment. Furthermore, the simplicity and compactness of the enriching inlet allow for potential applications with certain low-cost sensors to improve their detection capabilities in a wide range of environmental and medical sampling.

## Appendix A: Average enrichment for all experimental conditions tested

**Table A1.** Average enrichment offered by a 325 cm length of enriching inlet for all compounds at all operating conditions tested.

| Analyte | Flow (Q sccm) | Total pressure differential applied (dP bar) | Operating condition | Enrichment (%) Sub-atmospheric | Enrichment (%) High Pressure |
|---|---|---|---|---|---|
| methane | 2 | 2 | SA, HP | 70±3 | 40±1 |
| butane | 2 | 2 | SA, HP | 233±4 | 194±3 |
| pentane | 2 | 2 | SA, HP | 221±7 | 168±15 |
| $\alpha$-pinene | 2 | 2 | SA, HP | 304±6 | 296±9 |
| isoprene | 2 | 2 | SA, HP | 222±2 | 163±10 |
| cyclohexane | 0.2 | 3 | SA | 4640±414 | |
| | 0.5 | 3 | SA | 1870±12 | |
| | 1 | 2 | SA, HP | 751±12 | 695±31 |
| | 1 | 3 | SA | 1050±23 | |
| | 2 | 2 | SA, HP | 312±5 | 292±7 |
| | 3 | 1 | SA, HP | 115±1 | 111±4 |
| | 5 | 3 | SA† | 187±3 | |
| propane | 0.2 | 3 | SA | 3050±56 | |
| | 0.5 | 3 | SA, HP* | 1340±3 | 1120±8 |
| | 1 | 1 | SA, HP | 290±1 | 230±1 |
| | 1 | 2 | SA, HP | 559±1 | 470±8 |
| | 1 | 3 | SA, HP* | 773±3 | 651±39 |
| | 2 | 2 | SA, HP | 264±1 | 214±1 |
| | 5 | 3 | SA† | 237±1 | |

SA: Sub-atmospheric condition; HP: High pressure conditions; * averaged over n=2; † required extrapolation.

 **Appendix B:  Enrichment at ppb to sub-ppm analyte concentrations**

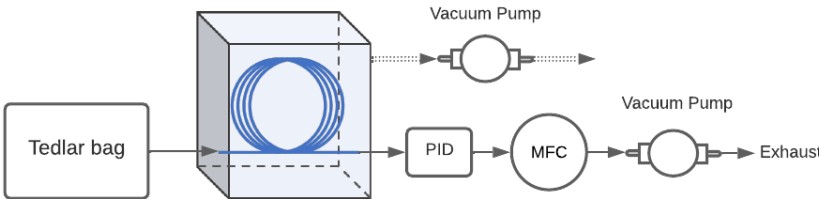

**Figure B1.** Schematic of the experimental setup used to confirm mutual independence of analyte concentration and enrichment offered by the inlet. The Teflon^TM AF-2400 is represented by the blue line, the miniPID2 photoionization detector is denoted by PID, and the mass flow controller downstream of the PID is denoted by MFC.

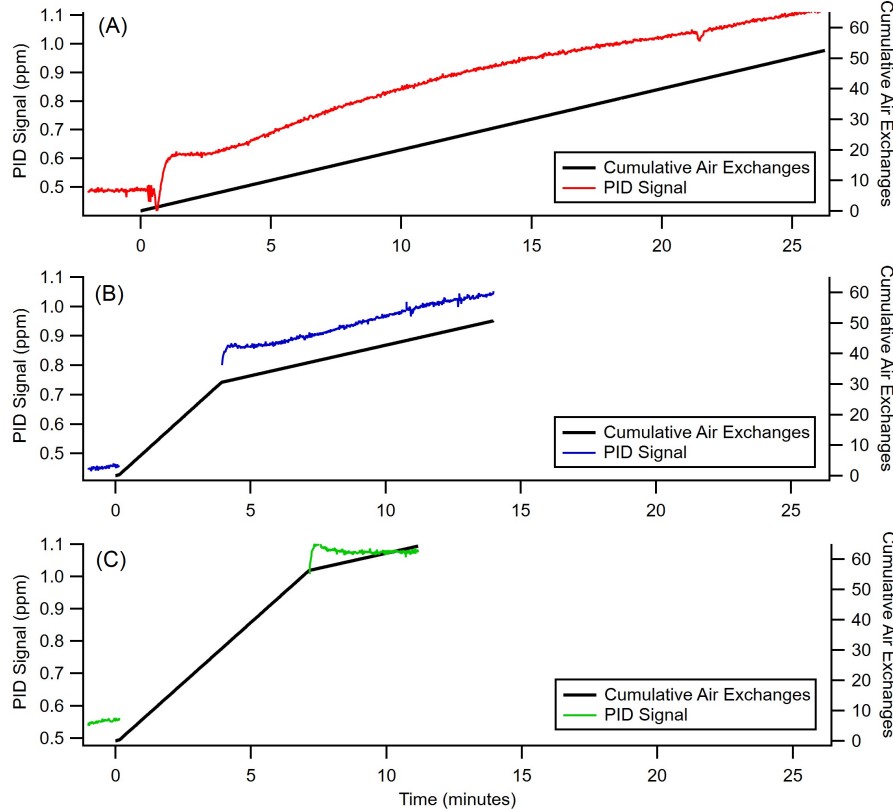

**Figure B2.** Time taken to reach equilibrium for enrichment of 500 ppb benzene ($C_6H_6$) at 2 sccm sampling flow rate, a pressure differential of 0.72 bar, inlet length of 300 cm. The primary y-axis shows the PID signal in ppm as a function of time and the secondary y-axis shows the cumulative air exchanges for (A) constant 2 sccm sampling flow, (B) 3 minutes at 8 sccm and remainder at 2 sccm sampling flow, (B) 6.5 minutes at 8 sccm and remainder at 2 sccm sampling flow.

*Author contributions.* **Namrata Shanmukh Panji:** Conceptualization, Methodology, Investigation, Writing - Original Draft, Formal analysis, Visualization **Gabriel Isaacman-VanWertz:** Conceptualization, Methodology, Writing - Review & Editing, Supervision, Funding acquisition

*Competing interests.* The authors declare no competing interests.

*Acknowledgements.* This work was supported in part by the National Science Foundation (AGS 1837882 and AGS 1837891) and the Environmental Protection Agency. Namrata Shanmukh Panji was also supported by the Edna Bailey Sussman Fund during this work. This article was developed under Assistance Agreement No. 84042501 awarded by the U.S. Environmental Protection Agency to Virginia Tech. It has not been formally reviewed by EPA. The views expressed in this document are solely those of the authors and do not necessarily reflect those of the Agency. EPA does not endorse any products or commercial services mentioned in this publication.

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
