# Peer review of "A Novel Inlet for Enriching Concentrations of Reactive Organic Gases in Low Sampling Flows"

_Atmospheric Measurement Techniques, 2023_

## Author Comment (AC1)

We thank the reviewers for their time and effort in evaluating our research and provide responses to comments and questions below. Reviewer comments are in black and our responses in blue, with new text shown in *italics*. Updated line numbers are noted for each comment.

General comments

Panji and Isaacman-VanWertz present a novel inlet that enriches concentrations of organic gases, allowing for detection of trace gases by low-cost sensors. Not only is this a generally helpful method but the presented material, equations, and experiment setup is straightforward and can easily be replicated, understood, and modified. My comments are mostly minor, with slightly more moderate comments about applicability to on-line measurements. I recommend publication after these comments are addressed.

Specific comments

Line 25-26: Can you elaborate on this a little more? Does the "majority of the analytical instruments" include these low-cost sensors or are you referring to the more expensive ones in the preceding sentences?

We have clarified that we are referring to both research-grade and low-cost instruments that have limits of detections in the ppb range. The text has been changed to read more clearly as follows:

L25-27: "Consequently, although many of the instruments currently used offer sensitivities *in the ppb range*, preconcentration steps are required to address the VOC levels in ambient samples below these limits of detection (Michulec et al., 2005)"

Line 35-36: This sentence, as written, does not tell us much. What are the major findings/limitations of this reference relative to what you are presenting here? It would be helpful to set up what advancements you are making for preconcentration of on-line sampling.

We agree with the reviewer that additional information would be useful and have sought to clarify our meaning in the revised text. Ras et. al (2009) describes techniques currently used to preconcentrate VOC samples for on-line or semi-online analytical methods. One of the methods that offers real-time preconcentration is the membrane introduction mass spectrometer (MIMS). However, techniques such as the MIMS are rather complete instruments which contain a component that preconcentrates sample flows and by introducing a simple inlet as in our work, we plan to lay the groundwork for a preconcentration technique compatible with other detectors (such as detectors designed for use with GCs) and future development in this area.

Additional information has been added to the text as follows:

L35-39: "As for preconcentration for on-line VOC detection, *membrane introduction mass spectrometers (MIMS) are currently being employed and improved upon (Ras et al., 2009). However, techniques such as the MIMS are rather complete instruments which contain a*

*component that preconcentrates sample flows. By introducing a simple inlet as in our work, we plan to lay the groundwork for a preconcentration technique adaptable with other detectors such as those compatible with GCs.*"

Line 134: What is this airtight material, its dimensions, the sub-atmospheric pressure you get it down to, and the pump you used?

We have clarified the text to read as:

L141-144: "The enriching inlet rested in an *airtight aluminum container (approximately 15cm x 10cm x 7.5cm) that was either open to the atmosphere (high pressure testing conditions) or closed and reduced to 0.2 bar absolute (sub-atmospheric testing conditions) by connecting to a diaphragm vacuum pump (GAST Manufacturing)*."

Line 163-165: Based on the given reference you should be able to estimate this change in permeation rate, right? Can you provide that here for your molecules?

The reviewer raises an interesting point. We note that the sub-atmospheric vs high pressure experiments conducted by Alentiev et al. (2002) were performed on a different variant of the tetrafluoroethylene (TFE) and 2,2-bis(trifluoromethyl)-4,5-difluoro-1,3-dioxole (BDD) polymer called AF-1600, though we expected similar patterns in the AF-2400 used here. The original permeability testing on AF-2400 was carried out by Merkel et al. (1999). Polymers AF-1600 and AF-2400 differ in their casting protocol and degrees of polymerization (0.65 and 0.87 respectively). As can be seen in Figure 8 of Alentiev et al. (2002), the permeabilities of molecules through AF-1600 and AF-2400 follow similar trends but differ by roughly one order of magnitude. For AF-1600, the sub-atmospheric permeabilities are roughly 50-60% (or rather 30-100 Barrer lower) of the permeabilities in high pressure experiments. Consequently, we expect qualitatively similar results for AF-2400 (i.e., lower permeability under sub-atmospheric testing), but quantitative estimates would be quite uncertain.

Alentiev et al. (1997) investigated the permeability coefficients of different molecules through a TFE/BDD polymer with a degree of polymerization of 0.9 (i.e., similar to AF-2400) using mass spectrometric methods where the pressure downstream of the polymer film was $10^{-4}$ Torr (near vacuum). Although this polymer had a different casting protocol as the AF-2400 (Merkel et al., 1999), we can extrapolate the permeation rates of nitrogen, and oxygen to be 554 and 1140 Barrer respectively. Assuming permeability coefficient of air to be equal approximately to the mole-weighted mixture, air permeability under sub-atmosphere conditions is equal to ~677 Barrer, in contrast to ~600 estimated in the manuscript based on the high-pressure conditions from 2002 Alentiev et al. reference. Similarly, permeability at sub-atmospheric conditions for methane was reported then as 435 Barrer in contrast to our estimated value of 400 Barrer. The enrichment for methane, calculated using Equation 11 in the manuscript, for sub-atmospheric conditions is therefore theoretically 1.2 (i.e., from the ratio of $P_{air}$ - $P_{analyte}$ at different temperatures) times that

during high-pressure condition testing. In our experiments, we see that this number is 1.75 which is 30% higher than expected. This could be attributed to differences in polymer casting or the methods used to determine permeability coefficients and the errors associated with them. These results support our above statement that the trends are qualitatively explainable, but quantitative estimates are not reliable. Since the permeability coefficients are not known for other molecules tested in our study and there do not yet exist methods to estimate them for sub-atmospheric conditions, we would also hesitate to extrapolate these values to other analytes. The text in the manuscript has been edited to read as follows:

L182-188: "Furthermore, results at a pressure differential ($\Delta$P) are dependent on the absolute pressures used to achieve the differential. The pressure downstream of the permeable membrane affects the permeation rate (Alentiev et al., 2002; *Alentiev et al., 1997; Merkel et al., 1999)*, so analyte permeation rates are expected to be lower through the membrane when the downstream pressure is close to vacuum (sub-atmospheric operation). *Based on permeability coefficients available in literature for low downstream pressures (Alentiev et al., 1997) compared to high pressure conditions (Alentiev et al. 2002), changes in permeation rates under low downstream pressures are estimated to improve enrichment by ~20% for methane. However, while a trend of improved enrichment under sub-atmospheric conditions is expected, quantitative estimation of improvement (including this 20% estimate) is highly uncertain due to a lack of data on most analytes and differences between specific polymers.*"

Line 186-187: This is related to the above comment. This can be estimated or even measured. Can you provide an estimate to help put a range on your error?

We agree that it would be best to estimate this overprediction. However, there are a number of factors that could cause differences between the model and the measurements that are difficult to constrain. For example, as described in our response to the comment above, the changes in the permeability coefficients due to low downstream pressures manifest as differences in the enrichments and are captured by conducting experiments with sub-atmospheric and high pressure conditions and are difficult to quantitatively estimate due to a lack of data on specific polymer properties. Prompted by the reviewer, we explored the impact of testing conditions as a function of compound and see that for almost all compounds, the difference is within 40% (refer to Figure R1).

[Figure]

Figure R1. Relationship between permeability coefficients and the ratio of enrichment during sub-atmospheric operating conditions vs that during high pressure conditions expressed as percentage.

On the other hand, to put a range to the error of the assumed permeability coefficient, we would have to consider that the experiments to quantify the relationship between critical volume and permeability coefficient conducted by Merkel et al., (1999) (which have been used in this study) do not report an error range to their estimation. Hence, it would be difficult to estimate the error associated with calculating permeability coefficients from critical volume. Similarly, experiments performed by Alentiev et al. (2002) from which permeabilities are estimated were conducted at 35 °C, and temperature is expected to impact enhancement as a function of molecule size (see discussion of temperature dependence in response to comment below, Figure R3). Consequently, while the model is generally predictive of enhancement, some uncertainty or bias is unavoidable. We have revised the language in the manuscript:.

L212-218: "Furthermore, observed enrichment tends to somewhat exceed that predicted for a given flowrate and pressure differential (dashed lines), which may be due to uncertainty in the estimated permeability coefficient or may be due to complex permeation behavior caused by the inherent pressure drop along the length of the permeable tubing, which acts as a restriction due to its narrow diameter. *Deviations between modeled and observed enrichment are likely due in part to uncertainty in permeabilities, as literature values were measured at slightly different temperatures and pressures (Alentiev et al., 2002) than those used here and it is difficult to quantify the precise impact of these differences on permeability for most of the analytes measured. In addition, the relationship with critical volume used to estimate permeability introduces some uncertainty.*"

Figure 4: Are these the 1σ of triplicate experiments? If so, please state that in the caption.

The authors thank the reviewer for pointing this out. We have edited the captions for figures 4, 5, 6, and B2 to include "error bars of 1σ *over repeated trials*" to account for the experiments that were performed in duplicates.

Line 220-221: Why is this an appendix section and not just the last part of the results and discussion section? You set up your introduction to say that low-cost sensors have a higher need for preconcentration for on-line sampling and here is an example of that. I would suggest including it in the main text.

We thank the reviewer for the suggestion. The Experimental Setup and Data Analysis section of the manuscript has been divided into two subsections. The previous setup is now listed under "High Concentration Experiments" and the experiment description in Appendix B is now under a subsection titled "Low Concentration Experiments". The discussion of the results and in Appendix B has been moved to the Results and Discussion section of the manuscript under a subsection titled "Effect of analyte concentration".

Appendix B: In your introduction you say that low-cost sensors fall short in measuring in the ppt range but your setup for on-line, real-time sampling does not address that. At best you can achieve a 1 bar pressure differential assuming you are drawing in air at ambient conditions without a pressure drop and your container is at vacuum. This leads to ~100% enrichment for more trace VOC according to Table A1 and Figures 4 and 5 which won't be detected by your miniPID2 if average concentrations are hundreds of ppt. If you decrease the sample flowrate or increase the inlet length your latency in detection may be too long and impractical. Is there a calculated estimate you can provide for an increase in enrichment through heating? That would be helpful for others who want to modify this.

We thank the reviewer for pointing this out. We agree there are some limits to the method, but there are several ways to provide enhancement beyond 100%, and the method is still useful for atmospheric concentrations. Using a longer enriching inlet (e.g., 10 meters) and sample flows of 1 sccm, low permeability compounds such as many of those tested here could reach enrichments of ~700%. For example, propane in Table A1 shows enhancement of ~300% in the current configuration despite its moderate permeability, which could be increased by lengthening the inlet. These enhancements are sufficient to bring a compound of ~200 ppt closer to 1 ppb. In other words, limits of detection improve by a factor of 5 or so (plus or minus a few). This might not be wholly revolutionary, but it is a significant benefit. For certain analytes, pressurizing the sample stream is feasible (for example, passing ambient air through a pump or compressor is commonly used for some types of whole-air sampling); in these cases higher pressure differentials are possible and truly transformative enhancements could be achieved. The impact of temperature is an interesting one. It may improve enhancement, but it is dependent on the activation energy of penetration and the Arrhenius relationship intercept for the analyte of interest. Merkel et al., (1999) studied the effects of temperature on permeability coefficients for Teflon AF-2400 and reported that generally, the permeability coefficients for nitrogen, oxygen, methane, and ethane increased

with increasing temperature. An increase in temperature will produce an increase in enrichment as long as the permeability of air increases faster than that of the analyte. We have calculated this effect for these molecules and find that enrichment may be improved at somewhat higher temperatures, but the efficacy of this approach is strongly dependent on energy of penetration, which is not well constrained for larger molecules of atmospheric interest. Temperature consequently may improve enrichment, but testing this effect would require significant changes to the experimental setup for gains expected to be somewhat marginal. If the reviewer is interested further in the temperature effects. A more detailed discussion of this issue and a figure of the modeled temperature effect (Figure R3) are included in the response to Reviewer 2.

Line 232-235: This is related to the above comment, but I see the flowrate of sample into a detector as a major limitation of this method since this method works best with <2 sccm of sample flow. For many VOC-detecting instruments this is an issue since they normally introduce flow on the orders of hundreds of sccm and therefore you would need to dilute any enrichment you gain. Do you have any examples of instruments with sccm-level flowrates that would work with this method?

We thank the reviewer for this comment. We recognize that the low flows tested in this study deviate from the typical practices in VOC sampling and present inherent challenges. However, these low flows are similar to those used by GCs, and many low-cost sensors have a history as GC detector. With some engineering, these flows could potentially be integrated into personal air samplers or used to enrich samples for PIDs, as demonstrated in Appendix B. Further, we are working to incorporate this approach into a combined instrument that will be the focus of future work. The manuscript has been edited as:

L126-130: "Pressurized gas cylinders were used to deliver the sample to the enriching inlet through a pressure regulator at pressures between 1 and 3 bar absolute (14.5 to 43.5 psia) *and tests were performed at 0.2 sccm to 5 sccm. These low flows are aligned with typical GC flows and an application for future research is expected to incorporate this approach into a combined instrument that would allow some separation of analytes in ambient atmospheres provided some engineering and analytical challenges could be overcome."*

Line 249: There is not a large change in enrichment with concentration but there is a notable change. Do you have an explanation for why increasing concentration increases enrichment? If these changes are all within error, can you state that and what the error range is based on the assumptions you mention throughout the text?

We believe this is actually mostly experimental errors. Ion Science Ltd. reports that the ppb miniPID2 used in this study has a sensitivity of >30mV/ppm, range of >40 ppm, and minimum detection limit of 1 ppb. The response factor reported for benzene for this sensor is 0.5, so sensitivity is on the order of 0.015 mV/ppb. At sensitivities nearing μV/ ppb, we are prone to noise from electrical interferences from the surroundings of the detector. Hence,

test cases at 45 ppb are near the limits of the detector which could explain why it might appear that enrichment varies with concentration. The text in the manuscript has been changed to clarify this as follows:

L254-258: *"We observe that the concentration of the analyte does not significantly affect the level of enrichment. Though some apparent effect is observed, measurements at lower concentrations here are near this instrument's limit of detection and subject to some measurement uncertainty. Concentrations in these experiments are orders of magnitude lower than those in other experiments but still approximately follow predicted values even at these low concentrations, suggesting little impact of concentration if any."*

Line 250-252: What is meant by testing at ambient conditions? Isn't that what this test was representative of? Or do you mean in conditions where the concentrations may fluctuate on a shorter timescale?

The authors thank the reviewer for seeking clarification. We have edited the manuscript to include more information as follows:

L259-263: "*This, combined with the fact that residence times can be reduced by temporarily using high sampling flows (refer to Figure B2), suggests that transient changes in the atmospheric concentrations would be detectable with the enriching inlet but it would need to be tested in real-world ambient environments. Once further advancements are made to transform the preconcentration inlet into a fully operational instrument, tests can be conducted under real-life conditions to assess its performance.*"

Technical corrections

We sincerely thank the reviewer for their keen eye in identifying these errors. They have been addressed as described below.

Line 11-12: I suggest changing "man-made" to "human-made".

L11-12: "A wide range of volatile organic compounds (VOCs) are emitted through natural processes (biogenic emissions) and by *human*-made processes (anthropogenic emissions)"

Line 68: "… so it is an ideal…"?

L71: "Furthermore, Teflon is a preferred material for sampling atmospheric gases (Deming et al., 2019), so *it* is an ideal material for this application."

Line 73: can you briefly explain this unit conversion to get 3.35 x 10-11?

We agree Barrer is a bit of a confusing unit. On line 51 the manuscript notes the equivalence between $10^{-10}\frac{cm^3_{STP}\cdot cm}{cm^2\cdot s\cdot cmHg}$ and $3.35\times10^{-11}\frac{mole\cdot cm}{m^2\cdot s\cdot bar}$, demonstrated here:

$$1\ Barrer\ =\ 10^{-10}\ \frac{cm^3_{STP}\cdot cm}{cm^2\cdot s\cdot cmHg}$$

$$1\ Barrer\ =\ 10^{-10}\ \frac{cm^3_{STP}\cdot cm}{cm^2\cdot s\cdot cmHg}\times\frac{1\ mole}{22400\ cm^3_{STP}}\times\frac{0.01\ m}{1\ cm}\times\frac{1\ cm^2}{10^{-4}\ m^2}\times\frac{1cmHg}{0.01333\ bar}$$

$$1\ Barrer\ =\ 3.35\times10^{-11}\frac{mole\cdot cm}{m^2\cdot s\cdot bar}$$

Figure 6: Some labels are boxed. You should revise to remove the boxes.

This seems to be an artifact of the LATEX to pdf conversion due to transparent image backgrounds rather than the boxes being present on the original figures. Affected figures have been changed to non-transparent backgrounds to address this.

Figure B2: In the last line of the caption "benzene" is misspelled.

Figure 7 caption: "The gray dashed line represents the theoretical enrichment expected for *benzene* ($C_6H_6$) with a permeability coefficient of 20 Barrer."

The label for pentane in Figure 6 was $C_5H_{10}$ instead of $C_5H_{12}$. This has been corrected.

References:

Michulec, M., Wardencki, W., Partyka, M., & Namieśnik, J. (2005). Analytical techniques used in monitoring of atmospheric air pollutants. *Critical Reviews in Analytical Chemistry*, *35*(2), 117-133.

Ras, M. R., Borrull, F., & Marcé, R. M. (2009). Sampling and preconcentration techniques for determination of volatile organic compounds in air samples. *TrAC Trends in Analytical Chemistry*, *28*(3), 347-361.

Merkel, T. C., Bondar, V., Nagai, K., Freeman, B. D., & Yampolskii, Y. P. (1999). Gas sorption, diffusion, and permeation in poly (2, 2-bis (trifluoromethyl)-4, 5-difluoro-1, 3-dioxole-co-tetrafluoroethylene). *Macromolecules*, *32*(25), 8427-8440.

Alentiev, A. Y., Yampolskii, Y. P., Shantarovich, V. P., Nemser, S. M., & Plate, N. A. (1997). High transport parameters and free volume of perfluorodioxole copolymers. *Journal of Membrane Science*, *126*(1), 123-132.

Alentiev, A. Y., Shantarovich, V. P., Merkel, T. C., Bondar, V. I., Freeman, B. D., & Yampolskii, Y. P. (2002). Gas and vapor sorption, permeation, and diffusion in glassy amorphous Teflon AF1600. *Macromolecules*, *35*(25), 9513-9522.

We thank the reviewer for their thorough and insightful consideration of our manuscript. Please find responses to questions and comments below.

RC2:
The study by Panji and Isaacman-VanWertz presents an innovative technique for enriching organic gases, utilizing a commercially available pre-concentration inlet system. The authors have theoretically characterised and validated the inlet design parameters, operating conditions, and inlet efficiency, thereby enabling the predictable enrichment of a majority of atmospheric gases. The approach is intriguing and valuable, with the results demonstrating its potential as a pre-concentration system for low-cost sensors. The paper is well-articulated, and the methodology is clearly described. However, I believe that any study introducing a new sampling approach must demonstrate its applicability with a real-life example, a section that is currently missing from this manuscript. I recommend acceptance of the paper upon addressing the below comments.

General comment

A real-world application of this method is crucial for a technical study of this nature. A simple demonstration, such as a diel cycle of ambient air sampling, could significantly enhance the paper's impact and convince readers of the method's practicality.

Thank you for taking the time to review our study. We agree that an application to ambient sampling (such as the diel measurement suggested) would demonstrate the utility of the current approach. However, we note that applying this technique to ambient measurements would introduce a number of significant engineering challenges in order to yield meaningful demonstration data. A coupling between this inlet and a detector with very low dead volume needs to be engineered, and ambient data would represent a combination of many different analytes at once, making any observed variability difficult to interpret. Consequently, we focus the present manuscript on the fundamental principle behind the design and demonstrating that it works using laboratory samples that can provide interpretable quantification of the efficacy. Now that the operation of the inlet is better understood through the tests in this manuscript, this technology can be applied to ambient instrumentation in future work once engineering challenges are overcome.

Specific comments

L7. Could you elaborate on the choice of such low flow rates for your experiments? While the enrichment is evidently more efficient within this range, these low flows are not common in VOC sampling and in fact, they can be challenging to work with. Additionally, it would be beneficial if you could discuss the uncertainties associated with measuring flows at these low rates.

The authors appreciate the reviewer's attention to this aspect and have clarified below.

We agree that the low flows tested in this study are not common in VOC sampling and can be challenging to work with (see, for instance, our note about the engineering challenges above). A goal of this technology is to reduce the need to pre-concentrate or use high flows. We envision its use with sensors that have been shown to be effective as gas chromatography detectors (e.g., photoionization detectors), which have already been designed to operate at these low flows, rather than samples collected in canisters or sorbent tubes, which typically have higher flows.

With regards to uncertainty, in this work we use Alicat MCS-series mass flow controllers, accurate to 0.2% of the full scale (i.e., ±0.02 sccm using 10 sccm controllers as in this work). The lowest measured flow of 0.2 sccm therefore has uncertainty of roughly 10%, while all other flows have reported uncertainties more in the range of 1-3%; this may somewhat underestimate uncertainty as the operating conditions may not precisely mirror those for which the MFC was calibrated by Alicat, but it is reasonable to estimate that flows are generally measured in this work with less than 10% uncertainty. The manuscript has been edited to read as follows:

L138-141: "Two mass flow controllers (Alicat Scientific, Inc.) were used to measure flow into the tubing, $Q_{in}$, (MCS-100sccm, MFM in Figure 2) and control flow to the detector $Q_{out}$ (MCS-10sccm, MFC in Figure 2). *Accuracy of flows is reported by the manufacturer as 0.2% of full scale plus 0.8% of reading; most flows in this work are measured at 2-20% of full scale, indicating uncertainty of less than 10% even at low reported flow rates.*"

L26-35. You may want to add the cryogenic trapping in empty stainless steel tubes, which would however require very low temperatures and the use of liquid nitrogen (Apel et. al. 2016, Bourtsoukidis et al., 2017)

We thank the reviewer for providing helpful citations to strengthen our manuscript. The citations have been added further on in the manuscript as:

L28-31: "Cryogenic trapping, which involves passing the sample through a cold tube with glass beads *or silica granules (Bourtsoukidis et al., 2017)*, is a well-established method used for preconcentration of VOCs in atmospheric samples but presents issues with ice formations clogging sample flows (Oliver et al.,1996; Wang and Austin, 2006)."

L114-115. Ambient air is typically characterized by a complex atmospheric mixture, particularly at low mixing ratios. Could you provide more details on how you plan to separate individual compounds in such a complex mixture? This information is essential for understanding the real-world applicability of the system.

The reviewer raises an important issue. As noted here and in our response to the reviewer's general comment, a major complexity in ambient samples is the mixture of many different

confounding analytes. The operating flows of the enriching inlet are closely in line with the flow rates of typical gas chromatographs, so we envision an instrument that relies on some form of chromatographic separation. In the next phases of our research, we intend to use a short GC column to separate ambient samples after it passes through the enriching inlet. However, this involves overcoming some interesting and complex analytical challenges, so we do not seek to include it in the scope of this manuscript.

L126-130: "Pressurized gas cylinders were used to deliver the sample to the enriching inlet through a pressure regulator at pressures between 1 and 3 bar absolute (14.5 to 43.5 psia) *and tests were performed at 0.2 sccm to 5 sccm. These low flows are aligned with typical GC flows and the focus of future research will be to incorporate this approach into a combined instrument that would allow some separation of analytes in ambient atmospheres provided some engineering and analytical challenges could be overcome.*"

L130-131 & L166-170. In your experiments, you've used pressurised air for sample delivery. Could you discuss the potential limitations this might pose for atmospheric sampling? Are there specific scenarios or applications where this method would be particularly useful or challenging? For instance, I understand that this method might be well-suited for canister analysis.

The authors are grateful to the reviewer for bringing attention to this matter and seeking clarification.

As noted by the reviewer, the enriching inlet developed in this work has been successfully tested for both pressurized and ambient samples. Whole air samplers have previously shown that many analytes of interest can be passed through a compressor or pump without significant losses (Lerner et al., 2017, doi:10.5194/amt-10-291-2017), which means a sampling scheme could pressurize the sample stream to provide high enrichments. Similarly, if the sample stream comes from a high pressure canister sample (e.g., 60 psia in the reference cited here), that would enable enrichments of 10x or higher at 1 sccm flows. Passing samples through a pump may lead to losses of lower volatility compounds, for which a pressure differential of 1 bar is likely the limit. This has been addressed in part as follows:

L194-196: "*We note that ambient-pressure sampling is limited to a 1 bar pressure differential, but many analytes of interest can be passed through a compressor or pump without significant losses (Lerner et al., 2017), suggesting that a pressurized sample stream could enable high enrichments*."

L153 & Figure 3. There seems to be a significant delay (approximately half an hour) before the desired enrichment is achieved. However, in L174 you state that the residence time was 1 minute at 1 sccm. Could you provide some insight into the reasons for this discrepancy? Is this delay a result of the changes in the experimental setup?

We thank the reviewer for the comment, which prompted us to perform more experiments to better constrain and attempt to reduce the equilibration time. Our initial tests as published in the original manuscript suggested short residence times at low concentration, but improved testing driven by this response has not reliably reproduced that result. Rather, the tests conducted with the FID and the miniPID2 setups both seem to have similar equilibration times of approximately 20-30 minutes. However, we have also determined that this equilibration time is not time dependent per se, but rather occurs after an approximately fixed number of air exchanges time, which can be expedited by increasing flow.

To evaluate possibilities for decreasing equilibration times, we conducted additional tests to determine the effect of purge flow through the inlet at a higher flow rate (8 sccm). Time was measured for 500 ppb benzene to reach enriched concentrations at 2 sccm when a pressure differential of 0.72 bar was introduced using the setup described in Figure B1 of the manuscript. We tested three scenarios: constant 2 sccm flow, 3 minutes of 8 sccm before switching back to 2 sccm, and 6.5 minutes of 8 sccm before switching back to 2 sccm.

[Figure]

Figure R2. Time taken to reach equilibrium for enrichment of 500 ppb benzene at 2 sccm sampling flow rate, a pressure differential of 0.72 bar, inlet length of 300 cm. The primary y-axis shows the PID signal in ppm as a function of time and the secondary y-axis shows the cumulative air exchanges for (A) constant 2 sccm sampling flow, (B) 3 minutes at 8 sccm and remainder at 2 sccm sampling flow, (B) 6.5 minutes at 8 sccm and remainder at 2 sccm sampling flow.

The results of this experiment are as shown in Figure R2, which shows the signal of the PID in response to the 500 ppb benzene sample stream from the time that a 0.72 bar pressure differential is introduced on the inlet (i.e., a vacuum pump is switched on); only signal collected when the sample flow is at 2 sccm is shown, with the gap in data representing the period of higher (8 sccm) sample flow. As can be seen in panel (A), it required 30 minutes for a constant flow of 2 sccm to approach the equilibrated enriched concentration, equivalent to ~60 air exchanges under these inlet dimensions (residence time of ~30 seconds). By increasing the flow rate to 8 sccm for 6.5 minutes, we can achieve roughly 52 air exchanges in 6.5 minutes, which is shown in panel (C) to achieve enriched equilibrium in this time. Hence, it is possible to decrease equilibration time by temporarily increasing sampling flow rate. We might be able to theoretically reduce it to 1 minute by temporarily sampling at 60 sccm for this set of operating conditions, though that was not tested due to limitations of the mass flow controller. The text in the manuscript has been changed to read as follows:

L220-223: "*We also note that the response time of the system can be reduced to a few minutes in contrast to the long response time observed in Figure 3 by temporarily increasing sampling flow rates (e.g., 8 sccm purge flow, see Figure B2).*"

Figure R2 was added to Appendix B.

L194. From the figures, it appears that this 'slightly higher' difference might be statistically significant. Please elaborate in further discussion.

We believe this can be attributed to experimental errors. Ion Science Ltd. reports that the ppb miniPID2 used in this study has a sensitivity of >30mV/ppm, range of >40 ppm, and minimum detection limit of 1 ppb. The response factor reported for benzene for this sensor is 0.5, so sensitivity is on the order of 0.015 mV/ppb. At sensitivities nearing µV/ ppb, we are prone to noise from electrical interferences from the surroundings of the detector. Hence, test cases at 45 ppb are near the limits of the detector which could explain why it might appear that enrichment varies with concentration. The text in the manuscript has been changed to clarify this as follows:

L254-258: "*We observe that the concentration of the analyte does not significantly affect the level of enrichment. Though some apparent effect is observed, measurements at lower concentrations here are near this instrument's limit of detection and subject to some measurement uncertainty. Concentrations in these experiments are orders of magnitude lower than those in other experiments but still approximately follow predicted values even at these low concentrations, suggesting little impact of concentration if any*"

Line 217: Given that Flame Ionization Detectors are highly sensitive with detection limits in the lower ppt range (down to 1-2ppt), the rationale for using such high concentrations in your experiments is not clear from this statement. While the demonstration of functionality is understood, the need to pre-concentrate such high concentrations could be better justified.

It's worth noting that low-cost sensors using alternative methods might be more practical and easier to use at these high concentrations.

We apologize for the confusion and seek to clarify here.

While FIDs are frequently used to measure concentrations down to ppt range, this is achieved by pre-concentration of high flows. Direct measurements by FID such as those conducted here require concentrations in the ppm range. For example, the FID produced by SRI, which is the model used in this work, is spec'd as being capable of detecting down to 1 ppm (https://www.srigc.com/home/product_detail/fid---flame-ionization-detector), which is a bit optimistic in our experience. Other major producers of FIDs report similar sensitivity. Agilent reports a sensitivity of several pgC/s (https://www.agilent.com/en/product/gas-chromatography/gc-detectors/flame-ionization-detector) which is roughly equivalent to 0.1 ppm of a small hydrocarbon at 1 sccm, similar to the level reported by Shimadzu (https://www.shimadzu.com/an/service-support/technical-support/analysis-basics/fundamentals/detector.html). The SRI FID was used in this work due to its compatibility with custom application and its design for low-sccm flows. Since the detection limit of the SRI FID is 1 ppm, high concentration samples were chosen to test the enriching inlet. Applications to low cost sensors are indeed an important goal of this work, but as mentioned will require overcoming certain engineering challenges.

The Experimental Set-up section of the manuscript has been edited to add clarifying information as follows:

L144-148: "A Flame Ionization Detector (FID, SRI Instruments) was used to detect the analyte concentration. *This detector was chosen due to its availability, robustness, and flexibly engineered design for the low flows used in this work. The minimum detection limit of this detector is 1 ppm, requiring sample concentrations in the ppm to percent levels for testing the enriching inlet.*"

Lines 234-235: The impact of temperature on the method's effectiveness seems to warrant at least more detailed discussion. Could you elaborate on how temperature variations, such as those occurring within a diel cycle, might affect the performance of your method? This is particularly important in understanding the method's robustness in real-world conditions.

The authors raise an interesting question that we explore here in some detail. We have included some additional discussion in the revised manuscript based on the response below.

In general, temperature dependence of enrichment depends on the temperature dependence of the permeability of both air and the analyte of interest. Merkel et al., (1999) studied the effects of temperature on permeability coefficients for Teflon AF-2400 and reported the activation energy of penetration (refer to Table 3 of Merkel et al., (1999)), which

is the slope of the Arrhenius relationship of ln(Permeability Coefficient) vs 1/Temperature. Generally, the permeability coefficients for nitrogen, oxygen, methane, and ethane increased with increasing temperature. The higher this activation energy, the faster the permeability coefficient increases with temperature. An increase in temperature will produce an increase in enrichment as long as the permeability of air increases faster than that of the analyte. Permeability coefficient at a given temperature, $P_T$, can be calculated for any given compound as $P_T = exp\left(ln(A) - \frac{E_p}{RT}\right)$ where R is the gas constant, T is the temperature in Kelvin, $E_p$ is the activation energy of permeation calculated as the slope of the measured Arrhenius relationship between permeability coefficients and inverse temperature (i.e., ln(P) vs 1/T), and ln(A) is a constant that represents the intercept of this relationship and can be calculated given measured permeability coefficient at a known temperature. Table R1 lists the values of $E_p$ reported by Merkel et al., (1999) and calculated values of ln(A) based on their reported values of permeability coefficients at T=35°C. The temperature-driven change in enrichment can then be calculated as the enrichment at the new temperature relative to that at reference temperature (25 °C). Since values of $E_p$ and permeability are only available for methane and ethane, we have calculated this effect for these molecules (see Figure R3). We also include a hypothetical analyte with low permeability (~10 Barrer at 35 °C; typical of atmospherically relevant VOCs) and varying $E_p$; the range of $E_p$ shown roughly brackets those expected for small atmospheric gases based on their Lennard-Jones diameters ($d_{LJ}$, Poling, B. E., & Prausnitz, J. M., 2001), which is linearly correlated as $E_p \cong 2.5(d_{LJ}^2-10)$ based on Figure 9a in Merkel et al. (1999).

Table R1: Permeability parameters for air, methane, and ethane from Merkel et al. (1999) for Teflon AF-2400. Reported values of $E_P$ and values of ln(A) calculated from reported permeability coefficient at T=35°C

| Molecule | $E_p$ (kJ/mol) | ln(A) |
|---|---|---|
| $O_2$ | 6.3 | 9.3 |
| $N_2$ | 8.9 | 9.6 |
| $CH_4$ | 9.8 | 9.8 |
| $C_2H_6$ | 21.2 | 13.6 |

[Figure]

Figure R3. Predicted increase in enrichment compared to enrichment at 25°C as a function of temperature for methane, ethane, and a hypothetical analyte ($P_{T=35C}$ = 10 Barrer at varying values of activation energy of penetration ($E_P$)).

The x-axis ranges from -30°C to 400°C as Teflon AF-1600 is reported to be structurally sound until 400°C (Resnick, P. R., & Buck, W. H., 2002). In general, enrichment may be improved at somewhat higher temperatures, but without a good constraint on $E_p$, which is lacking for larger molecules of atmospheric interest, it is possible to exceed the maximum enrichment and quickly reduce enrichment. An optimum might be found experimentally, but operating the enriching inlet at different temperatures would require significant changes to the experimental setup. At ambient temperatures, enrichment has relatively minor fluctuations across ambient temperature expected over the course of a day as shown in Figure R4.

[Figure]

Figure R4. Predicted increase in enrichment compared to enrichment at 25°C at ambient temperatures for methane, ethane, and a hypothetical analyte ($P_{T=35C}$ = 10 Barrer at varying values of activation energy of penetration ($E_P$)).

The text in the manuscript has been edited to read as follows:

L276-281: "Furthermore, due to the temperature dependence of permeation through Teflon AF-2400, there may be additional opportunities to modulate operating conditions to improve enrichment, *for instance by heating the enriching inlet. However, temperature impacts permeabilities of both air and an analyte of interest, leading to a complex impact on enrichment that is not well theoretically constrained due to a lack of data on the temperature dependence of analyte permeation (Merkel et al., 1999). The magnitude of this enhancement is generally not expected to be large, and ambient temperature cycles are not expected to drive significant changes in enrichment*."

References

Apel, E. (2016). *UCAR/NCAR: Earth Observing Laboratory, Trace Organic Gas Analyzer (TOGA) for HIAPER, UCAR/NCAR, Earth Observing Laboratory*. Tech. rep., https://doi. org/10.5065/D6DF6P9Q.

Bourtsoukidis, E., Helleis, F., Tomsche, L., Fischer, H., Hofmann, R., Lelieveld, J., & Williams, J. (2017). An aircraft gas chromatograph–mass spectrometer System for Organic Fast Identification Analysis (SOFIA): design, performance and a case study of Asian monsoon pollution outflow. *Atmospheric Measurement Techniques*, *10*(12), 5089-5105.

Resnick, P. R., & Buck, W. H. (2002). Teflon® AF: A family of amorphous fluoropolymers with extraordinary properties. In *Fluoropolymers 2: Properties* (pp. 25-33). Boston, MA: Springer US.

Poling, B. E., & Prausnitz, J. M. (2001). O'Connell JP. *The properties of gases and liquids*.

Lerner, B. M., Gilman, J. B., Aikin, K. C., Atlas, E. L., Goldan, P. D., Graus, M., ... & de Gouw, J. A. (2017). An improved, automated whole air sampler and gas chromatography mass spectrometry analysis system for volatile organic compounds in the atmosphere. *Atmospheric Measurement Techniques*, *10*(1), 291-313.

Merkel, T. C., Bondar, V., Nagai, K., Freeman, B. D., & Yampolskii, Y. P. (1999). Gas sorption, diffusion, and permeation in poly (2, 2-bis (trifluoromethyl)-4, 5-difluoro-1, 3-dioxole-co-tetrafluoroethylene). *Macromolecules*, *32*(25), 8427-8440.